# Nano-Hydroxyapatite as a Delivery System for Promoting Bone Regeneration In Vivo: A Systematic Review

**DOI:** 10.3390/nano11102569

**Published:** 2021-09-29

**Authors:** Anis Syauqina Mohd Zaffarin, Shiow-Fern Ng, Min Hwei Ng, Haniza Hassan, Ekram Alias

**Affiliations:** 1Department of Biochemistry, Faculty of Medicine, Universiti Kebangsaan Malaysia, Bandar Tun Razak 56000, W.P. Kuala Lumpur, Malaysia; anissyauqina96@gmail.com; 2Faculty of Pharmacy, Universiti Kebangsaan Malaysia, Kuala Lumpur 50300, W.P. Kuala Lumpur, Malaysia; nsfern@ukm.edu.my; 3Centre for Tissue Engineering and Regenerative Medicine, Universiti Kebangsaan Malaysia, Bandar Tun Razak 56000, W.P. Kuala Lumpur, Malaysia; angela@ppukm.ukm.edu.my; 4Department of Human Anatomy, Faculty of Medicine and Health Sciences, Universiti Putra Malaysia, Serdang 43400, Selangor, Malaysia

**Keywords:** nano-hydroxyapatite, bone regeneration, in vivo, drug delivery, bone morphogenetic protein, scaffold

## Abstract

Nano-hydroxyapatite (nHA) has been widely used as an orthopedic biomaterial and vehicle for drug delivery owing to its chemical and structural similarity to bone minerals. Several studies have demonstrated that nHA based biomaterials have a potential effect for bone regeneration with very minimal to no toxicity or inflammatory response. This systematic review aims to provide an appraisal of the effectiveness of nHA as a delivery system for bone regeneration and whether the conjugation of proteins, antibiotics, or other bioactive molecules to the nHA further enhances osteogenesis in vivo. Out of 282 articles obtained from the literature search, only 14 articles met the inclusion criteria for this review. These studies showed that nHA was able to induce bone regeneration in various animal models with large or critical-sized bone defects, open fracture, or methicillin-resistant *Staphylococcus aureus* (MRSA)-induced osteomyelitis. The conjugations of drugs or bioactive molecules such as bone-morphogenetic protein-2 (BMP-2), vancomycin, calcitriol, dexamethasone, and cisplatin were able to enhance the osteogenic property of nHA. Thus, nHA is a promising delivery system for a variety of compounds in promoting bone regeneration in vivo.

## 1. Introduction

Hydroxyapatite (HA) is a member of the apatite family (composed of calcium and phosphates) with a chemical formula Ca_10_(PO_4_)_6_(OH)_2_. It can be found naturally occurring in biological sources such as mammalian bones (e.g., bovine), marine sources (e.g., fish bones or scales), shells (e.g., seashells and eggshells), plants and algae, and minerals such as limestone [1]. HA can also be chemically synthesized using various techniques, depending on the desired chemical composition, shape, and particle size. A review by Sadat-Shojai et al. (2013) [2] has classified the methods employed in HA synthesis into five groups: wet methods, dry methods, high-temperature processes, extraction from natural sources, and a combination of these procedures. Wet methods offer advantages over the other methods; wet methods employ lower temperature and allow control of the morphology and particle size of HA during fabrication. There are six techniques commonly being applied for wet methods: chemical precipitation [3], hydrolysis [4], sol-gel [5], hydrothermal [6], emulsion [7], and sonochemical [8]. Among these, the chemical precipitation technique is the most frequently used because it is the easiest and most cost-effective [2].

Over the past few decades, HA, especially the nano-sized HA, has gained immense attention as an orthopedic biomaterial and coating for metallic implants owing to its excellent biocompatibility, bioactivity, and osteoconductivity [9]. Previous studies have demonstrated that nHA-based biomaterials were able to promote the bone repair process without any signs of inflammation [10,11]. Since the composition of HA is similar to bone minerals, it was anticipated that HA would provide a suitable surface for cell adhesion [12]. The application of nHA as a coating for metallic implants have been shown to further reduce the implant rejection rate and speed up the healing times, as they allow a controlled and rapid osseointegration between the bone and the surface of the implant [13].

In addition, nHA is also widely used as a vehicle for the delivery of various drugs or bioactive molecules [14], particularly in the field of orthopedics, because it promotes targeted delivery to the bone and a more controlled release of drugs [15,16]. nHA mimics the bone mineral properties and has good biodegradability, which allows drugs to bind and released slowly to the bone. nHA usually has a small particle size of less than 100 nm in diameter, which is similar to the microarchitecture of the osseous tissues. By virtue of this, nHA allows a more efficient internalization and provides greater surface area for the adsorption of cells, drugs or other bioactive molecules [17]. Several reviews have outlined the application of nHA for the delivery of drugs and/or other bioactive molecules, i.e., growth factors, antibiotics, and chemotherapeutic drugs [18,19]. However, to the best of our knowledge, the application of nHA alone or as a delivery system to promote bone regeneration in vivo has not been systematically reviewed to date. In this review, we critically analyzed different studies focusing on the application of nHA as a delivery system in promoting bone regeneration in vivo. In addition, the effects of the conjugated drugs or active pharmaceutical ingredients in enhancing the osteogenic properties of nHA are also discussed in this review.

## 2. Materials and Methods

The design of the current systematic review followed the Preferred Reporting Items for Systematic Reviews and Meta-Analysis (PRISMA). The primary records retrieved during the comprehensive literature search were pre-clinical in vivo studies that investigated the use of nHA as a delivery system to promote bone regeneration.

### 2.1. Search Strategies

A systematic search was conducted on 19 April 2021 on two databases: Scopus and Ebscohost. The search strategy involved a combination of keywords, including (“hydroxyapatite” AND “nano*”) AND (bone” OR “regenerat*” OR “form*” OR “osteo*”) AND (“delivery system” OR “delivery vehicle” OR “carrier”). Articles search for each database was limited to only English language. In addition, reference lists of relevant studies were searched for other potential records.

### 2.2. Eligibility Criteria

The questions addressed for this study was formulated using the PICOS (Population, Interventions, Comparisons, Outcomes, Study) model: (1) studies that use animal models with bone defect or representing bone disease (population); (2) studies evaluating nHA as a delivery system to promote bone regeneration (interventions); (3) studies have control in interventions, for example, untreated (blank) control and nHA treatment alone (comparisons); (4) studies reporting the effects of nHA as a delivery system for promoting bone regeneration (outcomes); and (5) only in vivo studies (study design). Studies from 2001 to 2021 were included in the review if they fulfilled the PICOS criteria.

Studies will be excluded from the review if any of these following exclusion criteria are met: (1) the nHA was used in dental applications; (2) the HA used was not nano-sized (as described in the articles); (3) the biomaterial used was not HA; (4) studies reporting on the fabrication and characterization of nHA as a delivery system only; (5) any of the outcome parameters analyzed was not on bone tissues; (6) studies did not represent the application of nHA for bone regeneration; (7) the nHA used in the study was not for delivery system; and (8) studies assessing toxicity only. Additionally, non-primary studies, case reports, proceedings abstracts, editorials, letters, comments to the editor, reviews, meta-analyses, and book chapters were excluded from the review.

### 2.3. Studies Selection and Data Extraction

All articles obtained from the two databases were imported to a reference manager to remove any duplicates. These records were then primarily screened through the title and abstract to identify studies that fall under the scope of the review and fulfil the inclusion criteria. Following the title and abstract screening, full-text articles were retrieved, screened, and reviewed by three authors (ASMZ, HH, and EA). Studies identified to have any of the exclusion criteria during full-text screening were be excluded. Relevant details (i.e., the treatments conjugated with nHA, animal models used, methodology, and main findings from the studies) were extracted and documented in the data extraction form. Any disagreement throughout the screening process was resolved through discussions between the three reviewers.

## 3. Results

### 3.1. Studies Selection

The initial database search resulted in a total of 282 records, where 102 records were obtained from the Ebscohost platform, while 180 records were obtained from Scopus. Duplicates were removed before the title and abstract screening, which resulted in a total of 257 records. A total of 155 records were excluded for various reasons. The majority of studies that were excluded during this process involved in vitro and studies that focused solely on the physicochemical characterization of nanoparticles (n = 63). One study that was designed for dental application was excluded because the focus of this review is on the application of HA in orthopedics. A total of 17 studies were excluded because nHA was employed solely as a biomaterial for bone regeneration rather than as a drug carrier system. Seven studies were excluded because the findings of these studies did not represent the application of nHA for bone regeneration. Among these studies, two focused mainly on the development of animal models for ectopic bone formation, one study explored the potential of nHA delivery system for tumour suppression, one study explored the antibacterial property of nHA delivery system, and two studies focused on the application of nano-hydroxyapatite in promoting cell adhesion and differentiation.

During the full-text screening process, 17 studies were excluded. Among these, seven studies were identified as ‘not for bone repair’ because no bone defect was induced in the animals. Three studies were excluded because the nHA employed was not the actual carrier for the delivery system. Another seven studies were excluded due to grouping issues, i.e., no negative control group (blank). The majority of the studies have a negative control (blank), where the group of animals remain untreated. A study by Krishnan et al. (2020) did not have the untreated control (blank); however, this article was included in the review because nHA was not presented in the control group (stimulan + vancomycin). This allowed us to assess the efficacy of nHA on bone regeneration by comparing the outcomes of nHA treatment groups against the control groups. At the end of the screening process, a total of 14 articles were considered eligible for this review (Figure 1).

### 3.2. Study Characteristics

The eligible articles included in this review were primary studies published between 2005 and 2021. Nine studies were performed using rats [20,21,22,23,24,25,26,27,28], four studies used rabbits [29,30,31,32], and one study used dogs [33] as animal models. The total number of animals used is 15–56 for rats, 16–45 for rabbits, and 8 for dogs. However, one of the studies included in this review did not mention the total number of animals involved [22].

Most animals had defects created either on the tibia [33], femur [21,25,31], or cranium [20,22,23,26,27,28,30,32] prior to implantation of treatments into the defect sites. Most of the studies used the critical-sized defects with a diameter ranging between 3 and 8 mm in rats, created by drilling circular transosseous defects [20,21,22,23,26,27,28]. A large-sized defect in dogs used in a study by Itoh et al. (2005) was 20 mm in diameter, whilst Su et al. (2013) induced a 26 × 5 × 3 mm^3^ defect in rabbits. A study by Raina et al. (2020) used a rat model of open fracture, which was developed by creating a transversal cut through the bone using a sagittal saw without removing any bone. Additionally, there were two studies that employed animal models of osteomyelitis. The animals were induced with methicillin-resistant *Staphylococcus aureus* (MRSA) bacteria infection on the bone, which led to bone defects [24,29]. In all studies, the treatment duration was between 4 and 24 weeks.

All studies evaluated the gross morphology of bone either by using micro-computerized tomography (micro-CT) scanning, multiscale spiral computerized tomography (MSCT), X-ray, or mammography. Histology, histomorphometry, or immunohistochemistry analysis using various staining methods were also performed to observe the areas of newly formed bone. Three studies assessed other bone parameters such as mechanical strength of the bone [25], calcium content [23], and solid-state nuclear magnetic resonance (ssNMR) and Raman analysis of the newly formed bone [28]. Details for the individual study characteristics and outcome parameters are summarized in Table 1 and Table 2.

### 3.3. Effect of nHA on Bone Regeneration In Vivo

Majority of the studies reviewed here reported that animals treated with nHA-based scaffolds, bone grafts, or hydrogel implants displayed effective bone repair compared to the untreated defects (Table 3, Table 4 and Table 5). Generally, the group of animals with nHA interventions had greater bone volume (BV), tissue volume (TV), percentage of bone volume over tissue volume (% BV/TV), and callus formation at the defect site compared to the untreated animals (blank control) [20,21,22,23,26,27,28,30,31,33]. The animals without nHA intervention in these studies either did not show any bone bridging [20,21,22,23,24,25,26,27,28,29,30,31,33] or did not survive throughout the study period [32].

### 3.4. Improvement in the Osteogenic Properties of nHA when Conjugated with Drugs or Other Bioactive Molecules

#### 3.4.1. Proteins

The majority of studies included in this review employed nHA as a carrier for bone morphogenetic protein (BMP) (Table 3). These studies demonstrated that the conjugation of nHA-based composites with BMP-2 further improved and accelerated the formation of new bones compared to the composites without BMP-2.

A study by Itoh et al. (2005) showed that defects implanted with BMP-2-loaded nHA composites contributed towards earlier callus formation and a higher percentage of new bone [33], while Tan et al. (2012) found a high percentage of %BV/TV and trabecular thickness (TbTh) [26] compared to defects implanted with nHA composites alone. In a study by Kim et al. (2008), the implantation of BMP-2-loaded poly(lactide-co-glycolide)/nHA composite suspended in fibrin gel (fibrin gel + PLGA/HA + BMP-2) induced mineralization of the newly formed bone as indicated by the high levels of calcium deposition compared to the group implanted with fibrin gel + PLGA/HA alone. Another study by Zhang et al. (2016) reported that defect implanted with nHA/collagen/poly L-lactic acid (nHAC/PLLA) scaffold containing P17-BMP-2 (a synthetic peptide derived from BMP-2 residues 32-48) at concentrations of 2 and 10 mg/g resulted in a noticeable bone union after 4 weeks of implantation. Interestingly, treatment with nHAC/PLLA/P17-BMP-2 scaffold (10 mg/g) had higher radiographic and histological scores compared to 2 mg/g [30].

Curtin et al. (2014) and Su et al. (2013) assessed the dual delivery of BMP-2 with vascular endothelial growth factor (VEGF) and basic fibroblast growth factor (bFGF), respectively. The dual delivery of VEGF and BMP-2 via nHA scaffold resulted in a more advanced and accelerated bone regeneration process, as indicated by the larger area of newly formed bone and more new blood vessels formation compared to the polyethyleneimine (PEI)-based scaffold and mixed PEI and nHA-based scaffold [20]. Similarly, the defects implanted with nHA scaffolds containing BMP-2 alone or both BMP-2 and bFGF showed higher new bone formation compared to the defects implanted with the nHA scaffold alone. However, scaffolds containing both BMP-2 and bFGF showed more mature new bones and more fibrous collagen formation compared to BMP-2 alone [32].

Meanwhile, Raina et al. (2020) and Teotia et al. (2017) investigated the co-delivery of BMP-2 and zoledronic acid (ZA) using either GM-based bone bandage [25] or nano-cement (NC) containing nHA and calcium sulphate [28]. Both studies reported that the delivery of ZA alone does not affect bone healing. However, when combined with BMP-2, there was an increased new bone formation [25,28], stronger newly formed callus [25] and a higher amount of calcium deposition and mineralized tissue [28]. Collectively, these studies suggested that the osteogenic property of nHA is enhanced when incorporated with BMP-2 alone or in combination with other bioactive molecules.

#### 3.4.2. Antibiotics

Only two studies reviewed here explored the application of nHA for the delivery of antibiotics (Table 4). Both studies employed nHA as a carrier for vancomycin for the treatment of MRSA-induced osteomyelitis [24,29]. These studies showed that nHA scaffolds containing vancomycin have excellent bactericidal and osteogenic properties.

Jiang et al. (2012) discovered that the group of rabbits implanted with nHA scaffold containing vancomycin successfully reconstructed the bone defect without any recurrence of infection. On the other hand, Krishnan et al. (2020) showed that scaffolds incorporated with vancomycin using encapsulation (SE-V) and absorption (SA-V) methods have better bactericidal and bone regeneration properties at a higher concentration of 15 wt% compared to 5 wt% [24]. There was no significant difference observed in the osteogenesis activity between the SE-V and SA-V treatment groups. Defects implanted with SE-V15 and SA-V15 showed significant bone union and a larger area of newly formed bone when compared to the control group (stimulan (calcium sulfate) + vancomycin). The control group simply treated the infection with no evidence of bone union [24]. Therefore, nHA can be a promising carrier for vancomycin to promote bone regeneration and prevent recurrent infection in osteomyelitis.

#### 3.4.3. Other Drugs or Bioactive Molecules

Four studies explored the application of nHA for the delivery of other drugs or bioactive molecules such as dexamethasone [22,27], calcitriol [21], and cisplatin [31] (Table 5). The conjugation of nHA-based scaffolds or hydrogel systems with these compounds were able to further improve the bone regeneration process.

Rats with critical-sized defects implanted with dexamethasone-loaded mesoporous silica-coated nHA scaffold (Dex + nHA-MS) [22] or HA scaffold (Dex/HA) [27] had higher new bone volume and more mature newly formed bone compared to the defects implanted with scaffold alone. Tavakoli-Darestani et al. (2014) reported that 80% of the defects healed by the end of the study period when implanted with Dex/HA (compared to only 60% of defects healed with HA alone). In another study, Hu et al. (2021) employed nHA as a carrier for calcitriol using a hydrogel system (Gel + HA-D + M). It was demonstrated that the incorporation of calcitriol further enhanced the bone regeneration ability of the hydrogel system [21]. The level of osteogenic markers (e.g., OCN and COL-1) expressed in defects implanted with calcitriol-loaded Gel + HA-D + M were higher compared to the unloaded Gel + HA-D + M system. Taken together, the incorporation of dexamethasone and calcitriol with nHA is beneficial for promoting bone regeneration.

Luo et al. (2019) have studied the conjugation of an anticancer drug, cisplatin (DPP), with a hydrogel system containing sodium alginate, chitosan, and surface-modified nHA (OSA–CS–PHA) [31]. The defects implanted with OSA–CS–PHA and OSA–CS–PHA–DPP both showed an improved bone regeneration process compared to the control group (OSA–CS–Borax). However, their effect was comparable with each other [31]. Hence, it was suggested that nHA might be a suitable carrier for anticancer drugs, especially for the treatment of bone cancers.

## 4. Discussion

This systematic review focuses on the application of nano-sized HA as a drug delivery vehicle for promoting bone regeneration. This small-sized nHA acts as an excellent drug delivery vehicle because it provides a greater surface area for drug adsorption, hence high drug loading capacity [34]. A previous study by Cheng et al. (2015) [34] demonstrated that ZA had a higher percentage of binding when incubated with nHA compared to the micro-HA with values of 92% and 43%, respectively. Although there is evidence that the use of the smaller size of HA as biomaterials was associated with an increased inflammatory response [35], a study by Zhou et al. (2018) [36] showed that the incorporation of bone-forming agents such as BMP-2 could prevent this deteriorating effect. Moreover, Zhou et al. (2018) also showed that BMP-2-loaded nanostructured HA had higher osteogenic activity in vivo compared to the BMP-2-loaded HA microspheres [36].

Rodent models are commonly employed for bone-related in vivo studies due to their low cost and easy handling. Most of the studies included in this review involved rat models with critical-sized defects for the evaluation of the bone regeneration process [20,22,23,25,26,27,28]. Rabbit models are used for large-size bone defects since it is difficult to be created in rodents [31,32]. Occasionally, dogs are also used to study bone regeneration, especially when involving weight-bearing sites [33]. Dogs have an advantage because their mature bone closely mimics human bone, which makes them mainly suitable for bone mechanical studies [37].

The nHA-containing bioactive compounds discussed here were directly implanted into the bone as scaffolds in the form of bone grafts, bone bandage, bioceramic, and cement [20,22,24,25,27,28,29,30,32,33] or implantable hydrogel systems [21,23,26,31]. nHA-based scaffolds are widely used due to their well-known excellent biodegradability, non-toxicity, and osseointegration (bone growth within a load-bearing implant) [38,39]. However, difficulty in controlling the pore size and porosity of nHA scaffolds may limit their application as a drug delivery system. These scaffolds have better mechanical characteristics and drug loading capacities when they are combined with chitosan, gelatin, or other polymers like PLGA and PLLA. On the other hand, hydrogel is a type of polymer scaffold composed of 3D polymer chains that contribute to its superior mechanical strength. Implantable hydrogels have several advantages, which include their ability to encapsulate bioactive molecules and absorbable and excellent integration with the surrounding tissues [40].

In all studies reviewed here, bone regeneration was assessed utilizing a variety of techniques, mainly micro-CT, MSCT, X-ray, mammography, histological staining, and histomorphometry analysis. While mammography and X-ray scan only allow 2D imaging [23,25,29,30,32,33], micro-CT and MSCT scans provide a more detailed 3D imaging of the bone with better resolution for the observation of bone microarchitecture and bone density [20,21,22,24,26,27,28,31]. The improvement in bone regeneration was reflected through the increased BV, TV,% BV/TV, TbTh, callus formation, bone union and reduced trabecular separation (TbSp). Meanwhile, for the histology and histomorphometry analysis, outcomes were evaluated depending on the types of staining used on the tissues: hematoxylin and eosin (H&E) [20,21,22,23,24,25,26,27,29,30,31,32,33] and Villanueva bone staining [33] were used to observe the new cells and tissues formation in decalcified and non-decalcified tissue sections, respectively; Masson’s trichrome staining [21,26,28,32] was used to examine the collagen fibres expression; and Alizarin red staining [28] was used to detect the calcium deposition in mineralized bone.

BMPs are a group of proteins that belong to the superfamily of transforming growth factor-β (TGF-β) [41]. These proteins were initially discovered for their osteoinductive properties [42]. Nowadays, BMPs are extensively studied in animal models of bone regeneration. For instance, several studies have reported that BMPs, especially BMP-2, were able to induce bone regeneration in animals with bone defects [43,44]. Nevertheless, the burst release of BMP-2 upon administration and its short half-life warrants the need for a carrier for optimal delivery of BMP-2. Several studies included in this review demonstrated that the conjugation of BMP-2 with nHA scaffolds or hydrogel systems resulted in an increased and accelerated bone regeneration process. The incorporation of BMP-2 into nHA-based scaffold resulted in a controlled and sustained release of BMP-2, hence, prolonged its biological activity to promote bone regeneration [45].

There are also studies that assessed the dual delivery of BMP-2 with other growth factors or drugs using nHA delivery system to further enhance the bone regeneration ability. For example, Curtin et al. (2014) have incorporated VEGF, a growth factor that induces vascularization, into BMP-2-loaded nHA scaffolds to accelerate the process of bone regeneration. Previous studies have reported that co-delivery of BMP-2 and VEGF yielded a better bone repair outcome compared to the delivery of a single growth factor [46,47]. VEGF promotes angiogenesis as well as facilitates the recruitment of mesenchymal stem cells (MSCs) and osteoprogenitor cells. It also acts synergistically with BMPs to enhance cell survival and promote bone mineralization [48]. Similarly, Su et al. (2013) conjugated BMP-2-loaded nHA composite with another growth factor, bFGF, to further promote the proliferation of bone cells and increase the level of osteocalcin. The nHA composite scaffold containing both BMP-2 and bFGF resulted in a better osteogenic effect compared to nHA composite containing BMP-2 alone [32]. A more recent study by Song et al. (2017) reported that bFGF improves the bone-forming ability of BMP-2 by upregulating the expression of BMP-2 and its receptor [49].

Other studies included in this review explored the combination of ZA with BMP-2 into nHA scaffolds [25,28]. ZA is an example of bisphosphonate and is mainly used in the treatment of osteoporosis [50]. ZA has an anti-resorptive property; it prevents bone resorption by suppressing osteoclast activity. Although ZA is well-known to benefit osteoporosis patients, long-term use of high doses of ZA is associated with impaired bone remodeling process [51,52]. Hence, co-delivery of ZA with bone-forming agents such as BMP-2 may prevent this undesirable effect. This was previously shown by Jing et al. (2016), where co-delivery of BMP-2 and ZA resulted in a more efficient bone formation in osteoporotic rabbits compared to the BMP-2 or ZA treatments alone [53].

Vancomycin is a classic example of antibiotics used for the treatment of MRSA-induced osteomyelitis. This MRSA infection causes inflammation in the bone that leads to significant bone defects. The treatments for MRSA-induced chronic osteomyelitis involve debridement of the infected bone and long-term systemic antibiotics administration to eradicate the infection, followed by bone grafting to repair the bone defects [54]. However, a prolonged systemic administration of vancomycin at a high concentration may impose negative implications of serious side effects and increased risk for bacterial resistance. The conjugation of vancomycin with a delivery vehicle such as nHA will provide effective management of the infected bone as it allows sustained-release and targeted delivery of vancomycin. A previous study demonstrated that vancomycin was efficiently loaded on the surface of nHA particles and was slowly released over a long period of time [55]. Furthermore, findings from the in vitro experiments in studies by Jiang et al. (2012) and Krishnan et al. (2020) confirmed that incorporation of vancomycin into nHA did not affect its antibacterial activity. Hence, the implantation of vancomycin-loaded nHA into rats with MRSA-induced osteomyelitis not only eradicated the infection but also allowed the bone repair processes to occur [24,29].

Other drugs or bioactive molecules used for bone repair in studies reviewed here are calcitriol and dexamethasone. Calcitriol is the active form of vitamin D3 and has been extensively studied for the treatment of post-menopausal osteoporosis [56,57]. An in vivo study by Liu et al. (2015) [58] reported that the implantation of an absorbable calcitriol-loaded collagen membrane scaffold into rat’s mandibular bone was able to accelerate new bone formation and promote bone maturation. Nonetheless, the short half-life of calcitriol may limit its therapeutic uses. To overcome this drawback, Hu et al. (2021) incorporated this vitamin into an injectable hydrogel system containing nHA for a more sustained release. On the other hand, dexamethasone is an example of corticosteroids that is indicated for a wide range of conditions associated with inflammation. Interestingly, the incorporation of dexamethasone into nHA based orthopedic biomaterials was found to improve the osteogenic property of nHA in vivo [22,27]. This is in accordance with findings from an in vitro study by Amjadian et al. (2016) [59], which demonstrated that dexamethasone-loaded nanofibrous composite scaffolds containing nHA were able to promote the osteogenic differentiation of mesenchymal stem cells.

One study included in this review assessed the incorporation of cisplatin with an nHA-based hydrogel system [31]. Cisplatin is a well-known chemotherapeutic drug and is effective against various cancers, including carcinomas, lymphomas, and germ cell tumors. The study by Luo et al. (2019) employed two animal models for different purposes: the mouse model to observe the anti-tumor effect and the rabbit model to evaluate the bone regeneration ability of the cisplatin-loaded hydrogel system. Different animal models were used because it was difficult to establish tumors in rabbits and create a large-size defect in mice. Although the incorporation of cisplatin with the nHA-based hydrogel system effectively suppressed tumor growth in the mice, there was no significant improvement in the osteogenic property of the system in rabbits seen [31].

Recent in vitro studies have revealed that the conjugation of nHA with norcantharidin, a chemotherapeutic agent, and curcumin was able to suppress osteosarcoma [60,61]. Dhatchayani et al. (2020) reported that the treatment of osteosarcoma MG63 cell line with selenite-substituted HA significantly reduced the cell viability, and the incorporation of curcumin with the selenite-substituted HA further enhanced this cytotoxic effect. Similarly, Huang et al. (2020) demonstrated the inhibition of osteosarcoma MG63 cell line and improvement in the cell viability of normal MC3T3-E1 cell line after treatment with strontium/chitosan/HA/norcantharidin composite.

Findings from all studies included in this review suggested that nHA was able to induce bone regeneration and incorporation of nHA with drugs or bioactive molecules yielded an enhanced bone regeneration process. However, our systematic review is only limited to in vivo studies. The inclusion of in vitro studies in future systematic reviews may provide a better understanding of the osteogenic property of nHA biomaterials, specifically on the osteoblasts (bone-forming cells) and osteoclasts (cells responsible for bone resorption). Furthermore, this review also included studies in which mixtures of polymers (e.g., PLGA and PEG) and nHA were used as the carriers for drugs. Future systematic reviews should evaluate studies that employ only nHA as a drug carrier. Additionally, this review is only able to suggest that nHA can be a suitable biomaterial to carry drugs or protein for promoting bone regeneration. A comprehensive evaluation of studies that also include another treatment group, i.e., animals treated with drugs or bioactive molecules alone, should be conducted in the near future. The comparison between the outcomes of nHA delivery system treatment groups and this additional group would provide an insight into whether nHA is a good delivery vehicle for the improvement of the therapeutic efficacy of drugs. Finally, all 14 studies reviewed used only one method of drug administration: direct implantation of nHA and drugs into the bone defect. It would be interesting to see the impact of the nHA delivery system on bone regeneration when administered through other routes of administration such as parenteral or the non-invasive oral route.

## 5. Conclusions

In summary, the majority of studies showed evidence that nHA promoted bone regeneration. However, in the context of treating osteomyelitis, the inclusion of antibiotics such as vancomycin is essential for bone regeneration, whereby the implantation of nHA alone does not induce bone regeneration due to recurrent infection. nHA as a drug carrier also does not inhibit the efficacy of the drug and may enhance bone regeneration. Structural analysis of the bone revealed that bone defects treated with nHA showed a greater BV, TV,% BV/TV, and callus formation compared to the untreated bone defects. Accordingly, histology and other histomorphometry analysis performed in these studies confirmed the effective bone regeneration process in animals treated with nHA compared to the animals with untreated defects. The conjugation of BMP-2, calcitriol, and dexamethasone to nHA resulted in an enhanced and accelerated bone regeneration process; however, this was not seen when cisplatin was conjugated to nHA. In conclusion, nHA may be proposed as a suitable carrier for the delivery of various drugs or bioactive molecules to promote bone regeneration in vivo.

## Figures and Tables

**Figure 1 nanomaterials-11-02569-f001:**
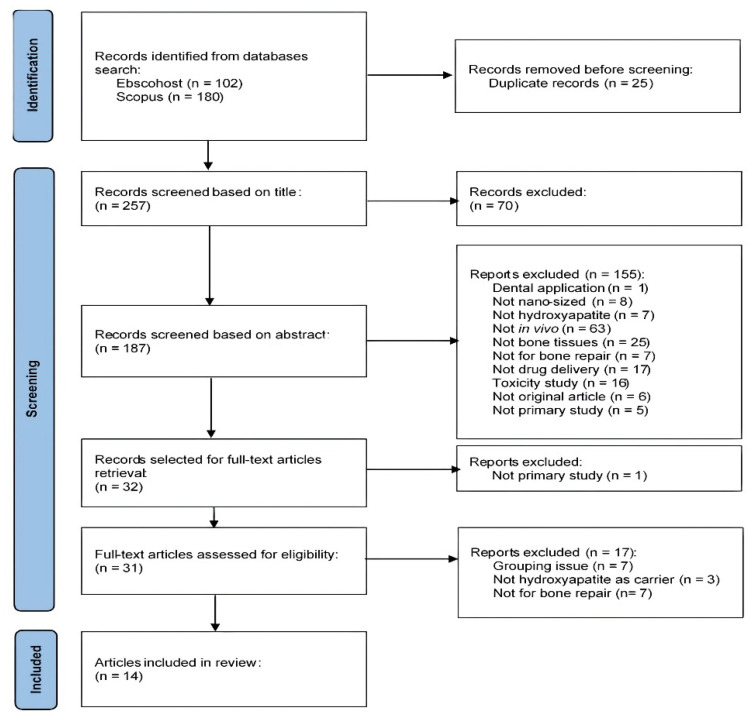
Flowchart for the identification and selection of studies according to the PRISMA statement.

**Table 1 nanomaterials-11-02569-t001:** Study characteristics.

Author (Year)	Animal Model	Bone Defect/Disease	Total No. of Animals	Post-Operative Observation Period
Curtin et al. (2015) [20]	Wistar rats	A 7 mm circular transosseous defect on the cranium	40	4 weeks
Hu et al. (2021) [21]	SD rats	A defect with 3 mm diameter on the femoral condyle of OVX rats	36	12 weeks
Itoh et al. (2005) [33]	Beagle dogs	A defect of size 20 mm on the central part of the tibia	8	12 and 24 weeks
Jia et al. (2021) [22]	SD rats	A defect of size approximately 8 mm diameter on the calvarial bone	Not mentioned	3 months
Jiang et al. (2012) [29]	NZ rabbits	MRSA-induced chronic osteomyelitis on the tibia	45	1, 2, 3, 6, and 12 weeks
Kim et al. (2008) [23]	SD rats	A critical size defect of 8 mm diameter on the parietal bone	24	8 weeks
Krishnan et al. (2020) [24]	Wistar rats	MRSA-induced osteomyelitis on the right femur	56	1 and 3 months
Zhang et al. (2016) [30]	NZ rabbits	A non-penetrating bone defect with a size of 10 × 5 × 5 mm^3^ on the mandibular bone	20	2 and 4 weeks
Luo et al. (2019) [31]	NZ rabbits	A defect on the femur	16	12 weeks
Raina et al. (2020) [25]	SD rats	An open fracture on the right femur	48	6 weeks
Su et al. (2013) [32]	NZ rabbits	A large-size defect of 26 × 5 × 3 mm^3^ on the mandible	36	4 and 12 weeks
Tan et al. (2012) [26]	SD rats	A critical-size defect of 8 mm diameter on the calvarial bone	18	4 and 8 weeks
Tavakoli-Darestani et al. (2014) [27]	SD rats	A critical-size defect of 8 mm diameter on the calvarial bone	15	8 weeks
Teotia et al. (2017) [28]	Wistar rats	A critical-size defect of 8.5 mm diameter on the calvarial bone	20	8 and 12 weeks

Abbreviations: SD—Sprague-Dawley; NZ—New Zealand; OVX—ovariectomized; MRSA—methicillin-resistant *Staphylococcus aureus.*

**Table 2 nanomaterials-11-02569-t002:** Outcome parameters.

Author (Year)	Micro-CT/X-ray/Mammography/MSCT (e.g., BV, TV, % BV/TV, Callus Formation, Bone Union)	Histology/Histomorphometry/Immunohistochemistry (e.g., Bone/Fibrous Tissue/Blood Vessel Formation, TbTh, TbSp, %Area of New Bone)	Mechanical Analysis (e.g., Mechanical Union/Non-Union, Peak Force, Extrinsic Stiffness	Other Specific Parameters
Curtin et al. (2015) [20]	+	+	-	-
Hu et al. (2021) [21]	+	+	-	-
Itoh et al. (2005) [33]	+	+	-	-
Jia et al. (2021) [22]	+	+	-	-
Jiang et al. (2012) [29]	+	+	-	-
Kim et al. (2008) [23]	+	+	-	+ (Calcium assay)
Krishnan et al. (2020) [24]	+	+	-	-
Zhang et al. (2016) [30]	+	+	-	-
Luo et al., (2019) [31]	+	+	-	-
Raina et al. (2020) [25]	+	+	+	-
Su et al. (2013) [32]	+	+	-	-
Tan et al. (2012) [26]	+	+	-	-
Tavakoli-Darestani et al. (2014) [27]	+	+	-	-
Teotia et al. (2017) [28]	+	+	-	+ (ssNMR and Raman analysis)

Abbreviations: Micro-CT—micro-computerized tomography; MSCT—multislice spiral computed tomography; BV—bone volume; TV—tissue volume; % BV/TV—percentage of BV over TV; TbTh—trabecular thickness; TbSp—trabecular separation; ssNMR—solid-state nuclear magnetic resonance.

**Table 3 nanomaterials-11-02569-t003:** Application of nHA for delivery of proteins.

Author (Year)	Interventions	Dosage	Delivery Approach	Significant Findings
Curtin et al. (2015) [20]	(1) Untreated (blank control) (2) Gene-free scaffold (3) PEI-pVEGF/PEI-pBMP-2 (PEI dual scaffold) (4) nHA/pVEGF/nHA-pBMP-2 (nHA dual scaffold) (5) PEI-pVEGF and nHA/pBMP-2 (mix dual scaffold)	* not specified in article	Implantable scaffold	nHA and combined PEI + nHA scaffolds containing both pBMP-2 and pVEGF showed higher new bone and vessels formation compared to the untreated defect
Itoh et al. (2005) [33]	(1) Untreated (blank control) (2) HAp/Col (3) HAp/Col + rhBMP-2 composite	(1) HAp/Col: HAp/Col only (2) HAp/Col + rhBMP-2: HAp/Col + 400 μg/mL rhBMP-2 solution	Implantable bone graft	Complete bone union was observed in both rhBMP-2 and non-rhBMP group, while the group with untreated defect does not show bone bridging throughout the study
Kim et al. (2008) [23]	(1) Fibrin gel only (blank control) (2) Fibrin gel + PLGA/HA (3) Fibrin gel + BMP-2 (4) Fibrin gel + PLGA/HA + BMP-2	(1) Fibrin gel + PLGA/HA: 0.1 mL Fibrin gel + 10 mg PLGA/HA (2) Fibrin gel + BMP-2: 0.1 mL Fibrin gel + 1 μg BMP-2 (3) Fibrin gel + PLGA/HA + BMP-2: 0.1 mL Fibrin gel + 10 mg PLGA/HA + 1 μg BMP-2	Implantable gel	An improved bone formation was observed in rats implanted with fibrin gels containing BMP-2 and PLGA/HA compared to the rats implanted with fibrin gel alone
Zhang et al. (2016) [30]	(1) Untreated (blank control) (2) nHAC/PLLA (3) nHAC/PLLA/P17-BMP-2 (2 mg/g) (4) nHAC/PLLA/P17-BMP-2 (10 mg/g)	(1) nHAC/PLLA/P17-BMP-2: nHAC/PLLA + 2 mg/g or 10 mg/g of P17-BMP-2	Implantable scaffold	Rabbits implanted with scaffold with or without P17-BMP-2 showed presence of new bone formation compared to blank control, which showed only small amount of callus formation
Raina et al. (2020) [25]	(1) Untreated (blank control) (2) GM (3) GM + ZA (4) GM + ZA + rhBMP-2	(1) GM: GM + 20 μL saline solution (2) GM + ZA: GM + 150 μg of ZA (concentration 4 mg/5 mL) mixed in 112.5 μL saline (3) GM + ZA + rhBMP-2: GM + 150 μg of ZA + 150 μg of rhBMP-2 (concentration 0.5 mg/mL)	Implantable bone bandage	The volumes of callus in all GM-treated groups were higher compared to the blank control, especially in the presence of rhBMP-2 and ZA
Su et al. (2013) [32]	(1) Untreated (blank control) (2) Scaffold only (3) BMSCs/scaffold (4) BMSCs/bFGF/scaffold (5) BMSCs/BMP-2/scaffold (6) BMSCs/bFGF/BMP-2/scaffold	(1) BMSC/scaffold: scaffold containing nHA + 1 × 10^7^ cells/implant (2) BMSCs/bFGF/scaffold: BMSCs/scaffold + 50 ng/mL bFGF (3) BMSCs/BMP-2/scaffold: BMSCs/scaffold + 100 ng/mL BMP-2 (4) BMSCs/bFGF/BMP-2/scaffold: BMSCs/scaffold + 50 ng/mL bFGF + 100 ng/mL BMP-2	Implantable scaffold	All rats implanted with scaffold showed areas of new bone formation compared to the untreated rats, where none of the rats survived
Tan et al. (2012) [26]	(1) PBS (blank control) (2) IBRC (3) IBRC/rhBMP-2	(1) IBRC: IBRC containing nHAC (2) IBRC/rhBMP-2: IBRC + 15 μg/mL of rhBMP-2	Injectable hydrogel system	Defects implanted with IBRC with or without rhBMP-2 showed new bone formation compared to the blank control, which did not show any bone repair
Teotia et al. (2017) [28]	(1) Blank control (2) NC (3) NC + ZA (4) NC + rhBMP-2 + ZA	(1) NC: cement containing nHA (2) NC + ZA: NC + 10 μg of ZA/disc (3) NC+ rhBMP-2 + ZA: NC + 2 μg of rhBMP-2/disc + 10 μg of ZA/disc	Implantable nano-cement	Defects implanted with NCs with or without ZA and rhBMP-2 showed new bone formation compared to the blank control, which did not show any new bone formation

Abbreviations: nHA—nano-hydroxyapatite; PEI—polyethyleneimine; pBMP-2,BMP-2—bone morphogenetic protein-2; pVEGF—vascular endothelial growth factor; BMSCs—bone marrow mesenchymal stem cells; HAp/Col—hydroxyapatite/collagen composite; rhBMP-2—recombinant human BMP-2; PLGA/HA—apatite-coated poly(D,L-lactide-co-glycolide)/nanohydroxyapatite particulates; nHAC/PLLA—nano-hydroxyapatite/collagen/polyL-lactic acid; P17-BMP-2—BMP-2-derived peptide; GM—gelatin-nano-hydroxyapatite membrane; ZA—zoledronic acid; BMSCs—bone marrow mesenchymal stem cells; bFGF—basic fibroblast growth factor; PBS—phosphate buffer solution (pH 7.4); IBRC—injectable bone regeneration composite; NC—nano-hydroxyapatite/calcium sulphate cement.

**Table 4 nanomaterials-11-02569-t004:** Application of nHA for delivery of antibiotics.

**Author (Year)**	**Interventions**	**Dosage**	**Delivery** **Approach**	**Significant Findings**
Jiang et al. (2012) [29]	(1) Blank group (2) nHA only (control group) (3) Vancomycin-loaded nHA (treatment group) * bones were debrided prior to implantation	(1) 160 mg of vancomycin/g nHA	Implantable pellets	Treatment group showed significant large areas of newly formed bone with no recurrent infection, while control group showed pus and new abscesses after 12 weeks
Krishnan et al. (2020) [24]	(1) Stimulan + vancomycin (2) Bare scaffold (3) SE-V5 (4) SE-V15 (5) SA-V5 (6) SA-V15 * bones were debrided prior to implantation	(1) Stimulan + Vancomycin: Stimulan (calcium sulfate) + 15 wt% vancomycin (2) SE-V: scaffold containing nHA + 5 wt% or 15 wt% vancomycin (3) SA-V: scaffold containing nHA + 5 wt% or 15 wt% vancomycin	Implantable scaffold	Scaffolds containing vancomycin (SE-V and SA-V) demonstrated good bactericidal and osteogenic properties compared to Stimulan + vancomycin, which only showed excellent bactericidal property without any new bone bridging

Abbreviations: nHA—nano-hydroxyapatite; SE-V—scaffold + entrapped vancomycin; SA-V—scaffold + absorbed vancomycin.

**Table 5 nanomaterials-11-02569-t005:** Application of nHA for delivery of other drugs or bioactive molecules.

**Author (Year)**	**Interventions**	**Dosage**	**Delivery Approach**	**Significant Findings**
Hu et al. (2021) [21]	(1) Gel (blank control) (2) Gel + HA (3) Gel + HA-D (4) Gel + HA-D + M (5) Gel + HA-D + Cal (6) Gel + HA-D + M + Cal	(1) HA-D + M: HA-D + 10 mg PCL-PEG-NH_2_ copolymers micelle (M) (2) HA-D + Cal: HA-D + 0.3 mg/mL Cal (3) HA-D + M + Cal: HA-D + M + 0.3 mg/mL Cal	Implantable hydrogel system	Hydrogel systems containing HA showed significantly higher new bone formation compared to blank control, especially in the presence of HA-D, M, and Cal
Jia et al. (2021) [22]	(1) Blank control (2) nHA (3) nHA-MS (4) Dex + nHA-MS	(1) nHA-MS: 0.2 g (2) Dex + nHA-MS: 20 mL Dex solution of 0.1 mg/mL + 0.2 g nHA-MS	Implantable scaffold	Defects implanted with nHA scaffolds have higher new bone formation compared to blank control, which barely showed any bone regeneration
Luo et al. (2019) [31]	(1) Blank control (2) OSA-CS-Borax (3) OSA-CS-PHA (4) OSA-CS-PHA-DDP	(1) OSA–CS–Borax: OSA + CS hydrogel + 0.02 g of Borax (2) OSA–CS–PHA: OSA + CS + 0.3 g of PHA (3) OSA–CS–PHA–DDP: OSA + CS + 0.3 g of PHA loaded with DDP	Implantable hydrogel system	The defect implanted with OSA–CS–PHA hydrogel showed large areas of new bone formation compared to the blank control and OSA–CS–Borax hydrogel implant
Tavakoli-Darestani et al. (2014) [27]	(1) Blank control (2) HA (3) Dex/HA	(1) Dex/HA: HA particles + 0.01 g/mL Dex	Implantable bioceramic	Defects implanted with HA bioceramics (with and without Dex) showed bone regeneration compared to blank control, which did not show any bone regeneration

Abbreviations: HA, nHA—nano-hydroxyapatite; HA-D—polydopamine coating nano-hydroxyapatite; PCL-PEG-NH_2_—amino-terminated poly(ethylene glycol)-block-poly(epsilon-caprolactone) copolymers; M—PCL-PEG-NH_2_ copolymers micelle; Cal—calcitriol; nHA-MS—nHA surface coated with mesoporous silica thin film; Dex—dexamethasone; OSA-CS—sodium alginate–chitosan; PHA—polydopamine-decorated nHA; DDP—cisplatin; Dex/HA—dexamethasone-loaded hydroxyapatite particles.

## Data Availability

The data presented in this study are available on request from the corresponding authors.

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
