# Peer review of "Nano-Hydroxyapatite as a Delivery System for Promoting Bone Regeneration In Vivo: A Systematic Review"

_nanomaterials, 2021, doi:10.3390/nano11102569_

Round 1

Reviewer 1 Report

The aim of the review is to evaluate the efficacy of nano-hydroxyapatite as a delivery system in promoting bone regeneration through the analysis of animal studies. The effectiveness of various molecules combined with nano-HA has been evaluated for the treatment of different conditions affecting bone.

The methodology of the review follows PRISMA guidelines, with the formulation of the research question according to PICO and it seems to be appropriate.

The manuscript appears well written and organized, however some points may be improved.

Line 50: a reference is missing

Line 66: a reference is missing

Line 82: maybe a search on the electronic database PubMed/MEDLINE may be recommended in order to detect a greater number of items that could be relevant for the focus of the review

Line 92: it could be useful to better highlight in this section what kind of comparison groups (controls) were requested for study eligibility

Line 310: a reference is missing

Conclusions: “The implantation of nano-HA alone did not induce bone regeneration due to recurrent infection” (L 432-433) seems in contrast with “all studies showed evidence that nano-HA promoted bone regeneration” (L 422). Please remove or clarify.

Reviewer 2 Report

The paper “Nano-hydroxyapatite as a Delivery System for promoting Bone Regeneration In Vivo: A Systematic Review” has been revised. This review focuses on the application of nano-sized HA as a drug delivery vehicle for promoting bone regeneration. This paper should be reconsidered after mayor revisión. The discussion of the results is very poor. Basically it limits itself to listing the results of the bibliography but does not explain how it affects the nHA and why?. Here are some examples: File 15, etc..:Change Nano-HA to nHA File 39: Enter reference number. File 290: Enter reference number. File 291: What is ZA? File 306-317: Osseointegration, biodegradability, and non-toxicity of nHA scaffolds should be fully explained. File 341-353: Explain the effect of dual delivery of BMP-2 with other growth factors and nHA. File 341-353: It should indicate that other growth factors have been used with BMP-2. File 354-367: Explain why vancomycin-loaded nHA eradicated the infection and also allowed the bone repair procecesses.

Round 2

Reviewer 2 Report

The author has reviewed the paper and made the proposed modifications. I think the paper can be published in the present form